# Clinical and Imaging Characteristics of Non-Neoplastic Spinal Lesions: A Comparative Study with Intramedullary Tumors

**DOI:** 10.3390/diagnostics12122969

**Published:** 2022-11-28

**Authors:** Keita Kajikawa, Narihito Nagoshi, Osahiko Tsuji, Satoshi Suzuki, Masahiro Ozaki, Yohei Takahashi, Mitsuru Yagi, Morio Matsumoto, Masaya Nakamura, Kota Watanabe

**Affiliations:** Department of Orthopaedic Surgery, Keio University School of Medicine, Tokyo 160-8582, Japan

**Keywords:** non-neoplastic intramedullary lesion, intramedullary tumor, magnetic resonance imaging (MRI)

## Abstract

The features of non-neoplastic lesions are often similar to those of intramedullary tumors, and a differential diagnosis is challenging to obtain in some cases. A surgical biopsy, which is performed on highly invasive tumors, should be avoided in cases of non-neoplastic lesions. Therefore, an accurate diagnosis is important prior to treatment. We evaluated 43 patients suspected of having spinal cord tumors and, finally, were diagnosed with non-neoplastic intramedullary lesions via magnetic resonance imaging. The patients commonly presented with myelitis. The patients with non-neoplastic neurological diseases had a significantly shorter symptom duration than those with intramedullary astrocytomas. The proportion of patients with non-neoplastic neurological diseases who presented with lesions at the cervical spinal level and focal lesions on axial imaging but without a spinal cord enlargement was significantly higher than that of patients with intramedullary astrocytomas. The current study aimed to distinguish spinal cord tumors from non-neoplastic intramedullary lesions based on their distinct features.

## 1. Introduction

An intramedullary spinal cord tumor is a rare pathological entity that commonly requires surgery [1,2,3]. It is primarily diagnosed via magnetic resonance imaging (MRI). However, in some cases, it is challenging to differentiate from non-neoplastic intramedullary lesions. Surgical treatment, including biopsy, is extremely invasive for non-neoplastic diseases, and it is associated with several risks, such as worsening symptoms and complications [4,5]. Therefore, obtaining an accurate diagnosis is extremely important before surgical intervention. However, the characteristics of non-neoplastic lesions, which are occasionally challenging to diagnosis in actual clinical practice, remain unclear.

To date, several studies have reported on the features of non-neoplastic spinal cord diseases. However, most of them are limited to case series or reports, which have a relatively small sample size [5,6,7,8,9,10]. Therefore, the clinical and imaging features of non-neoplastic intramedullary lesions are challenging to identify. To the best of our knowledge, only Cohen-Gadol et al. performed a descriptive study with a relatively large sample size of patients with neoplastic and non-neoplastic diseases who underwent spinal cord biopsy (*n* = 38). Nevertheless, a statistical analysis was not performed [4].

A recent study assessed the demographic characteristics and surgical outcomes of 67 patients with intramedullary spinal cord astrocytomas [11]. Non-neoplastic lesions on MRI frequently mimic astrocytomas [4,12,13]. Hence, it is reasonable to compare and validate the characteristic features of lesions to improve diagnostic accuracy.

The current study aimed to retrospectively evaluate the diagnosis and treatment of non-neoplastic diseases, which should be distinguished from intramedullary tumors, and to identify their pathogenesis.

## 2. Materials and Methods

### 2.1. Participants

This study was a single-center retrospective study. In our institute, neurologists are always consulted for cases that are suspected to be neurological diseases. All patients included in this current study underwent neurological examinations, and the correct diagnosis for neurological diseases was reached. Most of the cases with intramedullary tumors were diagnosed in our department (orthopedics), and very few cases were referred to neurologists.

In total, 43 consecutive patients who were referred to our department due to suspected intramedullary tumors and were ultimately diagnosed with non-neoplastic intramedullary lesions were included between 2001 and 2021. Patients diagnosed with vascular disorders, such as spinal arteriovenous fistula and spinal cord infarction, were excluded. Data on the diagnosis, demographic characteristics of the patients, MRI findings, and treatment were evaluated. We previously used 1.5-Tesla MRI, and, since 2015, 3-Tesla MRI was introduced and used. The cerebrospinal fluid (CSF) test was conducted by neurologists in 27/43 patients who had non-neoplastic spinal diseases. No patient underwent nerve conduction testing in this study.

In a subgroup analysis of neurological diseases, patients diagnosed with non-neoplastic intramedullary lesions were compared with patients (*n* = 67) diagnosed with intramedullary tumor (astrocytoma), as reported in a previous study [4].

### 2.2. MRI Findings

Lesional distribution was evaluated on axial T2-weighted MRI images. Based on the definition in previous studies [14,15], lesions with a hyperintensity area of ≥50% in the spinal cord were classified as extensive (A) and the other dot-shaped localized lesion as focal (B) (Figure 1).

Spinal cord enlargement was evaluated via sagittal T2-weighted MRI (Figure 2). 

N.N. evaluated the MRI findings of spinal cord enlargement and lesional area at two different time points. The intraobserver reliability tested with Cohen’s kappa coefficient was 0.93 for the enlargement and 0.85 for the lesional area (*p* < 0.01). All case images were evaluated independently by two spine surgeons (K.K. and N.N.) to assess the interobserver error. The interobserver reliability was 0.93 and 0.84 for the enlargement and lesional area, respectively (*p* < 0.01).

### 2.3. Statistical Analyses

Continuous variables and frequencies were presented as means ± standard deviations and categorical variables as percentages. The Student’s *t*-test and the chi-square test were used to compare the clinical characteristics between the groups. A *p* value of <0.05 was considered statistically significant. All statistical analyses were performed using the Statistical Package for the Social Sciences software version 28.0 (SPSS Inc., Chicago, IL, USA).

## 3. Results

### 3.1. Demographic Characteristics of the Participants

In total, 43 patients were included in the study. Among them, 23 (53.5%) were men and 20 (46.5%) were women. Their mean age was 43.7 ± 12.8 (range: 17–71) years (Table 1). The mean symptom duration at the initial visit was 7.4 ± 13.3 (range: 1–60) months. The initial symptom was limb and/or trunk numbness. In terms of other symptoms, 10 (23.3%) patients complained of limb weakness and 5 (11.6%) patients presented with bladder and/or bowel dysfunction.

The diagnoses were mainly divided into neurological diseases (*n* = 35) and cervical myelopathy (*n* = 8). Neurological diseases were diagnosed by neurologists. Myelitis of unknown etiology (*n* = 23 (53.5%)) was the most common neurological disease, followed by multiple sclerosis (*n* = 3 (7.0%)), neuromyelitis optica, and spinal cord sarcoidosis (*n* = 2 (4.7%)). Only one (2.3%) patient was diagnosed with atopic myelitis, autoimmune myelitis, HTLV-1-associated myelopathy, MOG antibody-associated disease, and Sjogren’s syndrome. In total, six (14.0%) patients were diagnosed with spondylosis and one (2.3%) patient with the ossification of the posterior longitudinal ligament and atlantoaxial subluxation (Table 1).

### 3.2. MRI Findings

Lesional location was evaluated via sagittal T2-weighted MRI, which showed hyperintensity intramedullary signals. A high-signal area was observed at the cervical and thoracic spinal levels in 32 (74.4%) and 8 (18.6%) patients, respectively. Three (7.0%) patients presented with multiple high-signal intensities in the whole spinal cord.

In total, 16 (37.2%) patients presented with spinal cord enlargement, and 27 (62.8%) did not. An extensive lesional area on axial imaging was observed in 13 (30.2%) patients, and 30 (69.8%) patients presented with focal lesions (Table 2).

The characteristics of symptom progression and imaging features are occasionally confusing because they are similar. Hence, the clinical features of neurological diseases were evaluated and compared with those of intramedullary astrocytomas. The results showed no significant differences in terms of age (41.8 ± 12.4 vs. 40.8 ± 18.6 years, *p* = 0.76) and sex (male: 48.6% vs. 56.7%, *p* = 0.43) (Table 3). However, patients with neurological diseases had a significantly shorter symptom duration than those with intramedullary tumors (4.8 ± 10.0 vs. 14.0 ± 19.1 months, *p* < 0.01).

In 35 patients who were diagnosed as having neurological diseases, 27 patients underwent a CSF examination, and 8 (22.9%) patients had both CSF pleocytosis and protein elevation. Two (5.7%) patients presented with CSF pleocytosis only, and another two (5.7%) patients showed protein elevation only. The other 15 patients did not show any specific characteristics through the CSF examination.

In terms of MRI features, there was a significantly high proportion of patients with neurological diseases who presented with lesions at the cervical spinal level. Meanwhile, most patients with intramedullary astrocytoma had lesions predominantly at the thoracic spinal level (*p* < 0.01). Next, we assessed the T1/T2 signal images and the extent of the contrast signal. In the T1-weighted signals, the lesions predominantly showed ISO signals in patients with neurological diseases and intramedullary astrocytomas. In contrast, T2 signals presented high signals at the lesion sites in both groups. In both groups, lesions were contrasted with gadolinium in the majority of cases, but there was no significant difference between the groups. When comparing the lesion size on T2 sagittal images, spinal neurological diseases were found to be significantly shorter along the longitudinal direction compared with intramedullary tumors (35.8 ± 43.9 vs. 76.5 ± 44.1 mm, *p* < 0.01).

All patients with intramedullary astrocytomas presented with spinal cord enlargement at the lesional level. Nevertheless, this condition was only observed in 40.0% (*n* = 14) of patients with neurological diseases (*p* < 0.01). In the lesional area on axial imaging, 25 (71.4%) patients with neurological diseases presented with focal lesions and 10 (28.6%) with extensive lesions. However, all patients with intramedullary astrocytomas had extensive lesions (*p* < 0.01).

The sensitivity and specificity findings of no spinal cord enlargement were 60.0% and 100% in neurological disease, respectively. Regarding the focal lesional area on axial imaging, the sensitivity and specificity were 71.4% and 100% in neurological disease, respectively.

### 3.3. Clinical Characteristics of Cervical Myelopathy

Eight patients were examined by both neurologists and orthopedics, and the diagnosis of neurological diseases was ruled out. The number of patients was limited. Hence, a statistical analysis was not performed. The patients’ mean age was 52.3 ± 12.1 (range: 34–71) years, and six (75.0%) were men. The mean symptom duration was 19.0 ± 20.0 (range: 1–60) months. Further, two of eight patients had slight spinal cord enlargement. Six patients presented with focal lesions, including a snake-eye appearance on axial imaging. Meanwhile, two patients had extensive lesions. Notably, in all cases, intramedullary high-signal lesions were observed on T2-weighted images, which was consistent with the level of intervertebral discs. Five patients underwent gadolinium-enhanced MRI, and two of five patients presented with enhanced lesions at the disc level.

### 3.4. Treatment

In total, 22 (51.2%) patients with neurological diseases received pulse steroid therapy and 13 (30.2%) were followed-up with conservatively. One patient diagnosed with Sjogren’s syndrome after surgical biopsy subsequently received steroid pulse therapy. Further, five (11.6%) of eight patients diagnosed with cervical myelopathy underwent surgery and three (7.0%) were followed-up with conservatively.

### 3.5. Representative Cases

Case 1: A 42-year-old woman who presented with acute onset upper and lower limb numbness was referred to our department. Gadolinium-enhanced cervical MRI at the initial visit showed a high-signal intensity at the C1/2 level. Moreover, spinal cord enlargement was detected. The lesion was diagnosed as atopic myelitis based on a detailed examination performed by neurologists, which revealed high blood IgE antibody levels. The patient received steroid pulse therapy and her symptoms improved. Three years after the therapy, the signal intensity disappeared (Figure 3).

Case 2: A 42-year-old woman who complained of upper and lower limb numbness was referred to our department. Gadolinium-enhanced MRI showed a high-signal intensity at the C6/7 level. The signal change was consistent with the intervertebral disc level. Neurologists performed a precise examination for the diagnosis of neurological diseases and all results were negative. The patient had mild myelopathy only, and she received conservative treatment with medications. After 6 years, the patient still presented with symptoms, which did not worsen. MRI showed a reduced intramedullary signal intensity. To date, a surgical intervention has not been planned (Figure 4).

## 4. Discussion

The clinical and imaging characteristics of non-neoplastic intramedullary lesions were evaluated. In the patients who were suspected of having an intramedullary tumor and referred to our department but who were finally diagnosed as having non-neoplastic spinal diseases, more than 80% was eventually diagnosed with neurological diseases. Meanwhile, the others were diagnosed with compressive cervical myelopathy, such as cervical spondylotic myelopathy (CSM) and the ossification of the posterior longitudinal ligament (OPLL). Patients with neurological diseases had a significantly shorter symptom duration than those with intramedullary astrocytomas. The proportion of patients with neurological diseases who presented with lesions at the cervical spinal level and focal lesions on axial imaging but without spinal cord enlargement was significantly higher than that of patients with intramedullary astrocytomas. Non-neoplastic intramedullary diseases have a heterogenous pathology, which could not be standardized. Therefore, this study emphasized their distinct features to distinguish them from spinal cord tumors.

Based on previous studies, patients with myelitis present with an acute onset of symptoms [16]. Myelitis has multiple causes, which include idiopathic, postinfectious, systemic inflammation, and multifocal central nervous system diseases [17]. Patients with neurological diseases included those with myelitis in our study. Hence, these patients had a significantly shorter symptom duration than those with neoplastic lesions. Similar to our results, Cohen-Godal et al. reported that, in their biopsy cases, inflammatory lesions had a shorter symptom duration than neoplastic lesions [4]. Thus, the clinical feature of disease duration could be a definite basis of neoplastic and neurological lesion diagnosis.

The use of spinal cord enlargement for the diagnosis of neurological lesions is controversial. Several studies have shown the absence of spinal cord enlargement in non-neoplastic lesions. However, others revealed an enlarged spinal cord [4,8,18]. Our results reported on the heterogenous characteristics of spinal cord expansion in non-neoplastic diseases (Table 2 and Table 3). However, notably, all astrocytoma cases had spinal cord enlargement in this study. This results indicated a higher possibility of diagnosing non-neoplastic lesions if there was no spinal cord expansion. Although there were rare cases of undetected spinal cord enlargement in patients with neoplasms [4], the lack of intramedullary expansion was a key finding for considering the diagnosis of non-neoplastic lesions.

Similar to the MRI findings including spinal cord enlargement, the T2-weighted axial images also showed 100% extensive lesions in patients with intramedullary astrocytomas (Table 3). This could be reasonable because intramedullary signal changes occur due to edemas, bleeding, syrinx formation, and cystic lesions, in addition to the tumor expansion itself [19,20]. In contrast, the transversely extensive lesions were limited to 28.6% in cases of neurological diseases, and most cases involved focal lesions (Table 3). This focal-type lesion is a characteristic finding of multiple sclerosis and certain types of myelitis based on the current and previous reports [15]. Therefore, a focal lesion at the axial section of MRI is a useful feature for diagnosing neurological diseases. However, diffuse and extensive lesions are observed, not only in astrocytomas, but also sarcoidosis, neuromyelitis optica, and other types of myelitis [12,21]. Hence, these lesions are occasionally difficult to distinguish from each other. In such cases, other distinguishing features, such as disease duration and intramedullary enlargement, could help obtain an accurate diagnosis.

Intramedullary astrocytomas are commonly located at the thoracic spinal level. Meanwhile, inflammatory spinal cord diseases are commonly located at the cervical spine [8,11,13,18]. This phenomenon was observed in our study, which revealed a frequent ratio of intramedullary lesions at the cervical spinal level in cases of neurological diseases (Table 3). However, these characteristics were not specific enough to obtain a differential diagnosis, and this criterion should only be used as a reference.

The diagnosis of cervical myelopathy is also confusing in some cases. Lee et al. reported on the existence of intramedullary signal intensity on T2-weighted MRI at the cervical intervertebral disc level [22], and similar results were observed in our study. If there is a signal change but canal stenosis is not evident, sagittal dynamic MRI could help obtain a diagnosis [23]. The signal intensity at the cervical disc level is a significant finding in cervical myelopathy. In some pathological conditions, the most compressed intramedullary area is enhanced with gadolinium [22,23,24,25,26]. The mechanism of this enhanced lesion could be attributed to cord substance disruption and the subsequent blood–spinal cord barrier breakdown, which promotes gadolinium uptake [27]. Even if this specific feature was identified, several patients underwent a biopsy or surgical resection of the enhanced area in previous reports, because MRI occasionally showed an abnormal extended high-signal intensity with cord enlargement, which mimicked intramedullary tumors [25]. Cord expansion might reflect edemas or a presyrinx state, which is induced by the obstruction of cerebrospinal fluid pathways at the stenotic spinal levels [22]. Thus, surgeons should keep these features in mind and conduct cautious examinations in collaboration with neurologists for the diagnosis of cord myelopathy.

The current study had several limitations. First, it was conducted from the perspective of spinal surgeons, and it aimed to distinguish neoplastic lesions from non-neoplastic diseases. Other studies should be conducted to obtain a detailed differential diagnosis of neurological diseases. Second, most patients in the current study did not undergo a biopsy or surgical resection. Therefore, an accurate diagnosis might not have been obtained in some pathologies, particularly idiopathic myelitis, which have no evident etiology. A neurologist could ultimately diagnose a patient with myelitis of unknown etiology, and similar cases were presented in this study. Third, we only showed cases of astrocytomas as a representative group of neoplastic lesions. An intramedullary tumor comprising different lesions and ependymoma is another common intramedullary tumor of the spinal cord [28,29,30]. The typical MRI characteristics of ependymoma are a central location, homogeneous enhancement, hemorrhage at the margins, and syringohydromyelia [12]. These are relatively distinguishable from non-neoplastic lesions. Therefore, ependymoma cases were excluded from the current study. Finally, the sample size was extremely small. Hence, a multivariate analysis could not be performed to eliminate confounders. Nevertheless, further studies should be conducted in the future to identify the independent distinctive features of non-neoplastic diseases with a larger sample size.

## 5. Conclusions

The characteristics of non-neoplastic spinal cord lesions were evaluated, and the patients commonly presented with myelitis. Hence, this condition should be differentiated from intramedullary neoplastic lesions. In terms of clinical features and MRI findings, compared with patients with intramedullary astrocytomas, those with non-neoplastic neurological diseases mainly presented with a shorter symptom duration, higher frequency of lesions at the cervical spinal level, no or slight spinal cord enlargement, and focal lesions in the transverse section. To prevent unnecessary surgical treatment, it is important to identify the characteristics of non-neoplastic spinal cord lesions and to consult neurologists to obtain an accurate diagnosis.

## Figures and Tables

**Figure 1 diagnostics-12-02969-f001:**
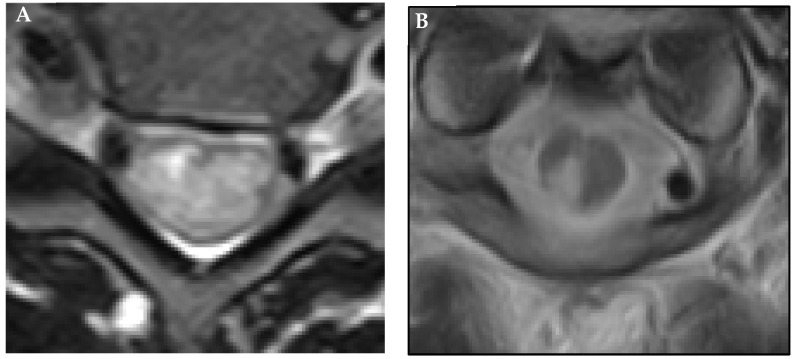
Lesional distribution on axial magnetic resonance imaging (MRI). Extensive (**A**) and focal (**B**) lesions showing a hyperintensity area on T2-weighted images.

**Figure 2 diagnostics-12-02969-f002:**
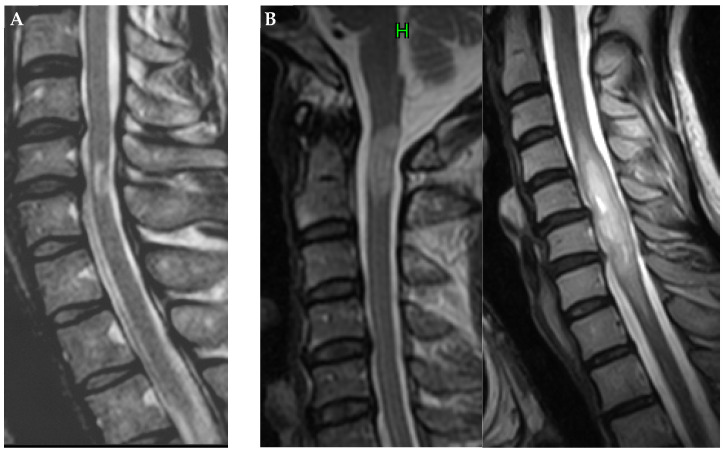
Sagittal T2-weighted MRI showing the absence (**A**) and presence (**B**) of spinal cord enlargement at the intramedullary hyperintensity area.

**Figure 3 diagnostics-12-02969-f003:**
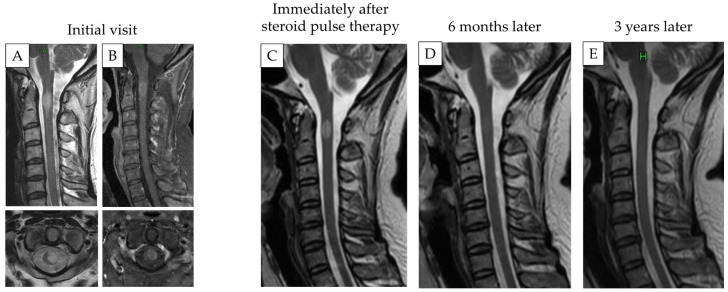
Representative MRI images of patients with atopic myelitis. Gadolinium-enhanced T2- (**A**) and T1-weighted MRI images (**B**) at the initial visit. Intramedullary signal change with cord enlargement was observed at the C1/2 level. (**C**–**E**) T2-weighted MRI immediately (**C**), 6 months (**D**), and 3 years (**E**) after steroid pulse therapy. The high-signal intensity disappeared over time.

**Figure 4 diagnostics-12-02969-f004:**
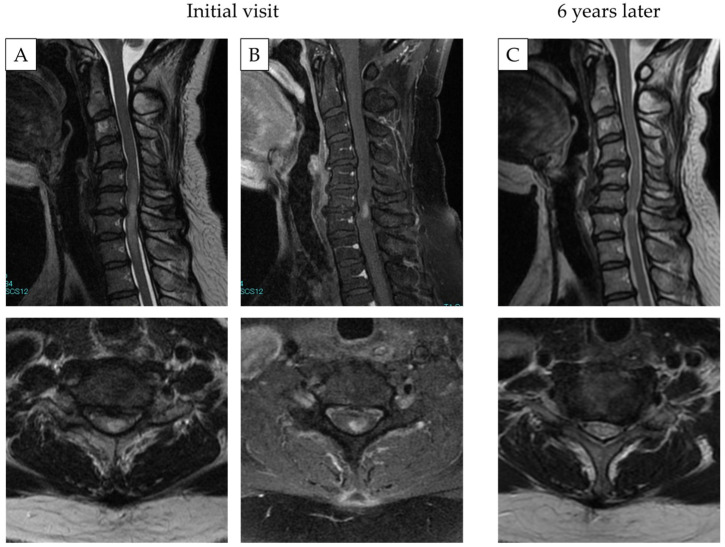
Representative MRI images of patient with cervical spondylotic myelopathy. Gadolinium-enhanced T2-(**A**) and T1-weighted MRI (**B**) at the initial visit. The lesion site was at the C6/7 level. (**C**) T2-weighted MRI after 6 years. Signal intensity was still noted.

**Table 1 diagnostics-12-02969-t001:** Demographic characteristics and diagnoses of 43 patients with non-neoplastic intramedullary lesions.

	Mean or Percentage
Age (years)	43.7 ± 12.8
Sex (male)	53.4
Duration of symptoms (months)	7.4 ± 13.3
Symptoms at the initial visit (%)	
Limb and/or trunk numbness	100
Limb muscle weakness	23.3
Bladder and bowel dysfunction	11.6
Diagnosis (%)	
Neurological disease	
Myelitis of unknown etiology	53.5
Multiple sclerosis	7
Neuromyelitis optica	4.7
Spinal cord sarcoidosis	4.7
Atopic myelitis	2.3
Autoimmune myelitis	2.3
HTLV-1-associated myelopathy	2.3
MOG antibody-associated disease	2.3
Sjogren’s syndrome	2.3
	2.3
Cervical myelopathy	
Spondylosis	14
OPLL	2.3
Atlantoaxial subluxation	2.3

HTLV-1: human T-cell leukemia virus type 1; MOG: myelin oligodendrocyte glycoprotein; OPLL: ossification of the posterior longitudinal ligament.

**Table 2 diagnostics-12-02969-t002:** MRI findings in 43 patients with non-neoplastic intramedullary lesions.

	Percentage
Lesional location (%)	
Cervical	74.4
Thoracic	18.6
Multiple	7
Spinal cord enlargement (%)	
No	62.8
Yes	37.2
Lesional area on axial imaging (%)	
Extensive	30.2
Focal	69.8

Characteristics of patients with neurological diseases and those with intramedullary tumors.

**Table 3 diagnostics-12-02969-t003:** Clinical characteristics and MRI findings between patients with neurological diseases and those with intramedullary astrocytomas.

	Patients with Neurological Disease (*n* = 35)	Patients with Intramedullary Astrocytomas (*n* = 67)	*p* Value
Age (years)	41.8 ± 12.4	40.8 ± 18.6	0.76
Sex (male)	48.6	56.7	0.43
Duration of symptoms (months)	4.8 ± 10.0	14.0 ± 19.1	<0.01
Lesional location (%)			<0.01
Cervical	68.6	41.8	
Thoracic	22.9	58.2	
Multiple	8.6	0	
Signal intensity (T1) (%)			0.18
Low	9.5	3.2	
High	2.4	0	
Iso	88.1	96.8	
Signal intensity (T2) (%)			-
Low	0	0	
High	100	100	
Iso	0	0	
Longitudinal length of hyperintensity signal (T2) (mm)	35.8 ± 43.9	76.5 ± 44.1	<0.01
Gd enhancement (%)			0.37
+	65.8	74.2	
−	34.2	25.8	
Spinal cord enlargement (%)			<0.01
No	60	0	
Yes	40	100	
Lesional area on axial imaging (%)			<0.01
Focal	71.4	0	
Extensive	28.6	100	

MRI: magnetic resonance imaging; Gd: gadolinium.

## Data Availability

The datasets generated and/or analyzed during the current study are available from the corresponding author on reasonable request.

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
