# Peer review of "Clinical and Imaging Characteristics of Non-Neoplastic Spinal Lesions: A Comparative Study with Intramedullary Tumors"

_diagnostics, 2022, doi:10.3390/diagnostics12122969_

Round 1

Reviewer 1 Report

Overall this is an interesting and important topic.  Often times distinguishing a neoplastic from non-neoplastic spinal cord lesion is of the utmost importance. This paper, while exploring some of the defining characteristics of non-neoplastic lesions, I feel could benefit from some further clarification.  First of all, could the authors expand on the various imaging characteristics such as contrast enhancement, T1 signal, and extent of T2/contrast signal? 

Author Response

We thank the reviewers for evaluating our manuscript entitledClinical and imaging characteristics of non-neoplastic spinal lesions: A comparative study with intramedullary tumors” (diagnostics-1945387).

We found the reviewers’ comments to be very helpful, and we have attempted to address the comments as thoroughly as possible. The reviewers’ comments are presented in bold, with our corresponding responses below. The changes made to the revised manuscript are highlighted in red.

Reviewer 1-1) First of all, could the authors expand on the various imaging characteristics such as contrast enhancement, T1 signal, and extent of T2/contrast signal? 

According to the reviewer’s feedback, we assessed the MRI and the extent of the T1, T2 and contrast signal. In the T1-weighted signals, the lesion predominantly showed iso signals in patients with both neurological disease and intramedullary astrocytomas. In contrast, T2 signals presented high signals at the lesion site in both groups. In both groups, lesions were contrasted with gadolinium in the majority of cases, but there were no significant differences between the groups.

When comparing the lesion size on T2 sagittal images, spinal cord tumors were found to be significantly larger along the longitudinal direction (35.8 ± 43.9 vs. 76.5 ± 44.1 mm, P < 0.01). These results indicate that when large lesions are observed, intramedullary tumors should be suspected first rather than neurological diseases. The Table 3 was revised as follows and the results are shown in the table below.

Table 3. Clinical characteristics and MRI findings between patients with neurological disease and those with intramedullary astrocytomas

Patients with neurological disease (n = 35)

Patients with intramedullary astrocytomas (n = 67)

P value

Age (years)

41.8 ± 12.4

40.8 ± 18.6

0.76

Sex (male)

48.6

56.7

0.43

Duration of symptoms (months)

4.8 ± 10.0

14.0 ± 19.1

<0.01

Lesional location (%)

Cervical

Thoracic

Multiple

68.6

22.9

8.6

41.8

58.2

0

<0.01

Signal intensity (T1) (%)

Low

High

Iso

9.5

2.4

88.1

3.2

0

96.8

0.18

Signal intensity (T2) (%)

Low

High

Iso

0

100

0

0

100

0

-

Longitudinal length of hyperintensity signal (T2) (mm)

35.8±43.9

76.5±44.1

<0.01

Gd enhancement (%)

+

-

65.8

34.2

74.2

25.8

0.37

Spinal cord enlargement (%)

No

 Yes

60.0

40.0

0

100

<0.01

Lesional area on axial imaging (%)

Focal

 Extensive

71.4

28.6

0

100

<0.01

MRI: magnetic resonance imaging, Gd: gadolinium.

(page 5, lines 172–179)

Next, we assessed the T1/T2 signal images and the extent of contrast signal. In the T1-weighted signals, the lesion predominantly showed iso signals in patients with neurological disease and intramedullary astrocytomas. In contrast, T2 signals presented high signals at the lesion site in both groups. In both groups, lesions were contrasted with gadolinium in the majority of cases, but there was no significant difference between the groups. When comparing the lesion size on T2 sagittal images, spinal neurological diseases were found to be significantly shorter along the longitudinal direction compared with the intramedullary tumor (35.8 ± 43.9 vs. 76.5 ± 44.1 mm, P < 0.01).

Please see the attachment. The changes made to the revised manuscript are highlighted in red.

Reviewer 2 Report

Dear authors,

The reviewer has read the paper and made some comments, which the reviewer hopes the authors will correct with reference to the paper. 

Summary

This article compared the clinical and imaging characteristics between the non-neoplastic intramedullary lesion and the astrocytoma. They found non-neoplastic lesions included more cervical spinal level and focal lesions on axial imaging without spinal cord enlargement compared to intramedullary astrocytomas. As the authors mentioned in the manuscript, it is essential to distinguish non-neoplastic lesions from neoplasms such as astrocytoma by clinical characteristics and imaging study. This study provided essential findings. The reviewer has some questions and requests before acceptance.

 ï¼‘.In the authors' institution, do patients with intramedullary lesions always consult neurologists? Please add this point to the discussion section.

 ï¼’.In this study, who did evaluate MRI? Please provide inter and intra-observer reliability of MRI findings of spinal cord enlargement and extensive axial lesion.

 ï¼“.In myelopathy patients, two patients showed spinal cord enlargement. However, the reviewer thinks it is difficult to evaluate spinal cord enlargement with spinal stenosis.

 ï¼”.In the discussion section, line 183-184, the authors described," More than 80% of patients suspected of intramedullary tumor and referred to our department were eventually diagnosed with neurological diseases." This sentence was difficult to understand. Did the authors include all patients with the intramedullary lesion, including tumors, and 80% of them were a neurological disease? The reviewer thinks that this sentence should be corrected.

5.In the discussion section, line 205-206, the authors described, "However, notably, there was no case of spinal cord enlargement in astrocytomas in this study." However, the reviewer thinks that this was finding about non-neoplasticity. Please confirm.

Author Response

Kajikawa et al.

Point-by-point responses to the reviewers’ comments

We thank the reviewers for evaluating our manuscript entitledClinical and imaging characteristics of non-neoplastic spinal lesions: A comparative study with intramedullary tumors” (diagnostics-1945387).

We found the reviewers’ comments to be very helpful, and we have attempted to address the comments as thoroughly as possible. The reviewers’ comments are presented in bold, with our corresponding responses below. The changes made to the revised manuscript are highlighted in red.

Reviewer 2-1) Using this criteria, it would be helpful to estimate the sensitivity and specificity of this method to summarise the takeaway message for clinicians. 

This proposal is of great significance. The sensitivity and specificity findings of no enlargement of the spinal cord were 60.0% and 100% in neurological disease, respectively. In terms of the focal lesional area on axial imaging, the sensitivity and specificity were 71.4% and 100% in neurological disease, respectively.

(page 5, lines 186–189)

The sensitivity and specificity findings of no spinal cord enlargement were 60.0% and 100% in neurological disease, respectively. Regarding the focal lesional area on axial imaging, the sensitivity and specificity were 71.4% and 100% in neurological disease, respectively.

Reviewer 2-2) Another thing to look at would be the lumbar puncture results and possibly nerve conduction study results as they can also help in the diagnosis of myelitis.

In 35 patients who were diagnosed as neurological diseases, 27 patients underwent the cerebrospinal fluid (CSF) test, and 8 (22.9%) patients had both CSF pleocytosis and protein elevation. Two (5.7%) patients presented CSF pleocytosis only and another 2 (5.7%) patients showed the protein elevation only. The other 15 patients did not show any specific characteristics through the CSF examination. The CSF test was not performed in all patients who were diagnosed with astrocytomas. No patient underwent nerve conduction testing in this study. The following sentence was added to both the Methods and Results section:

(page 2, lines 61–63)

The cerebrospinal fluid (CSF) test was conducted by neurologists with 27/43 patients who had non-neoplastic spinal diseases. No patient underwent nerve conduction testing in this study.

(page 5, lines 164-168)

In 35 patients who were diagnosed as neurological diseases, 27 patients underwent CSF examination, and 8 (22.9%) patients had both CSF pleocytosis and protein elevation. Two (5.7%) patients presented CSF pleocytosis only and another 2 (5.7%) patients showed the protein elevation only. The other 15 patients did not show any specific characteristics through the CSF examination.

Reviewer 2-3) For the technical side, it would be interesting to look at the limitations of the strength of the MRI machine (how many Teslas do the machine at your institute has?)

Although we previously used 1.5 Tesla MRI, 3 Tesla MRI was introduced and has been used since 2015. As the reviewer pointed out, the different Tesla MRIs could affect the diagnosis of spinal cord diseases. However, the correct diagnosis for all cases was eventually reached through either 1.5 or 3 Tesla MRI, which might not affect the diagnostic process. In fact, Hagens et al. reported that compared to 1.5 Tesla MRI, 3 Tesla MRI did not have any predominance to improve neurological disease diagnosis (Neurology, 2018). Therefore, we believe that the different strengths of MRI do not have an impact on the features of the image findings.

Reference

Hagens MH, Burggraaff J, Kilsdonk ID, Ruggieri S, Collorone S, Cortese R, Cawley N, Sbardella E, Andelova M, Amann M, Lieb JM, Pantano P, Lissenberg-Witte BI, Killestein J, Oreja-Guevara C, Wuerfel J, Ciccarelli O, Gasperini C, Lukas C, Rovira A, Barkhof F, Wattjes MP; MAGNIMS Study Group. Impact of 3 Tesla MRI on interobserver agreement in clinically isolated syndrome: A MAGNIMS multicentre study. Mult Scler. 2019 Mar;25(3):352-360.

(page 2, lines 60–61)

We previously used 1.5 Tesla MRI, and since 2015, 3 Tesla MRI has been introduced and used.

Please see the attachment. The changes made to the revised manuscript are highlighted in red.

Reviewer 3 Report

Dear Authors I appreciated the research article proposed.

The paper is of interest, and the significance of the content is relevant.

Anyway, I am sorry to say that the imaging (MRI) analysis is very superficial. I suggest You to re-analyse the MRI scans of patients taken into accounto all the relevant MRI features including: lesion dimensions, signal intensity (T1 and T2), contrast enhancement (when available), oedema of adjacent spinal chord, increasing / thickness of spinal chord, area of malacia ....

Then after a deepen analyses you can compare your results of non-neoplastic lesions with  those already reported in the literature of tumors . This would help to offer something useful in clinical practice. In another way, you could also create a control group of neoplastic patient and compare the MRI findings (but this is by far more complex and could maybe done in a future research).

I think that these analyses will significantly improve your paper

Author Response

Kajikawa et al.

Point-by-point responses to the reviewers’ comments

We thank the reviewers for evaluating our manuscript entitledClinical and imaging characteristics of non-neoplastic spinal lesions: A comparative study with intramedullary tumors” (diagnostics-1945387).

We found the reviewers’ comments to be very helpful, and we have attempted to address the comments as thoroughly as possible. The reviewers’ comments are presented in bold, with our corresponding responses below. The changes made to the revised manuscript are highlighted in red.

Reviewer 3-1) The paper is of interest, and the significance of the content is relevant. Anyway, I am sorry to say that the imaging (MRI) analysis is very superficial. I suggest You to re-analyse the MRI scans of patients taken into accounto all the relevant MRI features including: lesion dimensions, signal intensity (T1 and T2), contrast enhancement (when available), oedema of adjacent spinal chord, increasing / thickness of spinal chord, area of malacia ....

Then after a deepen analyses you can compare your results of non-neoplastic lesions with those already reported in the literature of tumors. This would help to offer something useful in clinical practice. In another way, you could also create a control group of neoplastic patient and compare the MRI findings (but this is by far more complex and could maybe done in a future research).
I think that these analyses will significantly improve your paper

I appreciate the reviewer’s valuable proposal. As Reviewer 1 pointed out, we assessed the MRI and the extent of the T1, T2 and contrast signal. In the T1-weighted signals, the lesion predominantly showed iso signals in patients with both neurological disease and intramedullary astrocytomas. In contrast, T2 signals presented high signals at the lesion site in both groups. In both groups, lesions were contrasted with gadolinium in the majority of cases, but there were no significant differences between the groups.

When comparing the lesion size on T2 sagittal images, spinal cord tumors were found to be significantly larger along the longitudinal direction (35.8 ± 43.9 vs. 76.5 ± 44.1 mm, P < 0.01). These results indicate that when large lesions are observed, intramedullary tumors should be suspected first rather than neurological diseases. The Table 3 was revised as follows and the results are shown in the table below.

Table 3. Clinical characteristics and MRI findings between patients with neurological disease and those with intramedullary astrocytomas

Patients with neurological disease (n = 35)

Patients with intramedullary astrocytomas (n = 67)

P value

Age (years)

41.8 ± 12.4

40.8 ± 18.6

0.76

Sex (male)

48.6

56.7

0.43

Duration of symptoms (months)

4.8 ± 10.0

14.0 ± 19.1

<0.01

Lesional location (%)

Cervical

Thoracic

Multiple

68.6

22.9

8.6

41.8

58.2

0

<0.01

Signal intensity (T1) (%)

Low

High

Iso

9.5

2.4

88.1

3.2

0

96.8

0.18

Signal intensity (T2) (%)

Low

High

Iso

0

100

0

0

100

0

-

Longitudinal length of hyperintensity signal (T2) (mm)

35.8±43.9

76.5±44.1

<0.01

Gd enhancement (%)

+

-

65.8

34.2

74.2

25.8

0.37

Spinal cord enlargement (%)

No

 Yes

60.0

40.0

0

100

<0.01

Lesional area on axial imaging (%)

Focal

 Extensive

71.4

28.6

0

100

<0.01

MRI: magnetic resonance imaging, Gd: gadolinium.

(page 5, lines 172–179)

Next, we assessed the T1/T2 signal images and the extent of contrast signal. In the T1-weighted signals, the lesion predominantly showed iso signals in patients with neurological disease and intramedullary astrocytomas. In contrast, T2 signals presented high signals at the lesion site in both groups. In both groups, lesions were contrasted with gadolinium in the majority of cases, but there was no significant difference between the groups. When comparing the lesion size on T2 sagittal images, spinal neurological diseases were found to be significantly shorter along the longitudinal direction compared with the intramedullary tumor (35.8 ± 43.9 vs. 76.5 ± 44.1 mm, P < 0.01).

The changes made to the revised manuscript are highlighted in red.

Reviewer 4 Report

Good paper trying to differentiate non-neoplastic from neoplastic lesions when a spinal cord lesion is seen. You looked at symptom duration, location (cervical), presence of spinal cord enlargement and focal lesions in the transverse section as your predictors.

Using this criteria, it would be helpful to estimate the sensitivity and specificity of this method to summarise the takeaway message for clinicians. 
Another thing to look at would be the lumbar puncture results and possibly nerve conduction study results as they can also help in the diagnosis of myelitis.

For the technical side, it would be interesting to look at the limitations of the strength of the MRI machine (how many Teslas do the machine at your institute has?)

Author Response

Kajikawa et al.

Point-by-point responses to the reviewers’ comments

We thank the reviewers for evaluating our manuscript entitledClinical and imaging characteristics of non-neoplastic spinal lesions: A comparative study with intramedullary tumors” (diagnostics-1945387).

We found the reviewers’ comments to be very helpful, and we have attempted to address the comments as thoroughly as possible. The reviewers’ comments are presented in bold, with our corresponding responses below. The changes made to the revised manuscript are highlighted in red.

Reviewer 4-1) In the authors' institution, do patients with intramedullary lesions always consult neurologists? Please add this point to the discussion section.

In our institute, neurologists are always consulted for cases that are suspected to be neurological diseases. All the patients who were included in the current study underwent neurologists’ examinations, and the correct diagnosis for neurological diseases was reached. Most of the cases with intramedullary tumors were diagnosed in our department (orthopedics), and very few cases were referred to neurologists. The following sentence was added to the Methods section:

(page 2, lines 49–54)

In our institute, neurologists are always consulted for cases that are suspected to be neurological diseases. All the patients included in this current study underwent neurologists’ examinations, and the correct diagnosis for neurological diseases was reached. Most of the cases with intramedullary tumors were diagnosed in our department (orthopedics), and very few cases were referred to neurologists.

Reviewer 4-2) In this study, who did evaluate MRI? Please provide inter and intra-observer reliability of MRI findings of spinal cord enlargement and extensive axial lesion.

N.N. evaluated the MRI findings of spinal cord enlargement and lesional area at two different time points. The intraobserver reliability tested by Cohen's Kappa coefficient was 0.93 for the enlargement and 0.85 for the lesional area (P < 0.01). All case images were evaluated independently by two spine surgeons (K.K. and N.N.) to assess the interobserver error. The interobserver reliability was 0.93 and 0.84 for the enlargement and lesional area, respectively (P < 0.01).

(page 3, lines 97–102)

N.N. evaluated the MRI findings of spinal cord enlargement and lesional area at two different time points. The intraobserver reliability tested by Cohen's Kappa coefficient was 0.93 for the enlargement and 0.85 for the lesional area (P < 0.01). All case images were evaluated independently by two spine surgeons (K.K. and N.N.) to assess the interobserver error. The interobserver reliability was 0.93 and 0.84 for the enlargement and lesional area, respectively (P < 0.01).

Reviewer 4-3) In myelopathy patients, two patients showed spinal cord enlargement. However, the reviewer thinks it is difficult to evaluate spinal cord enlargement with spinal stenosis.

The reviewer pointed out an important issue. Here is a representative case of a patient who was diagnosed with CSM and presented with spinal cord enlargement on the initial visit to our department, as shown in the figure below. Spinal cord enlargement was detected at the C6 level, adjacent to the most stenotic C6–C7 level. The C6 level enlargement was resolved 5 years after conservative treatment. The anteroposterior length of the spinal cord at the C6 level was 8.2 mm at the initial visit and 4.8 mm at the final follow‑up visit. Therefore, we believe that there were some cases that showed spinal cord enlargement, although they were initially diagnosed with cervical myelopathy.

Reviewer 4-4) In the discussion section, line 183-184, the authors described," More than 80% of patients suspected of intramedullary tumor and referred to our department were eventually diagnosed with neurological diseases." This sentence was difficult to understand. Did the authors include all patients with the intramedullary lesion, including tumors, and 80% of them were a neurological disease? The reviewer thinks that this sentence should be corrected.

We apologize if the sentence was not clear. In the current study, 43 consecutive patients who were referred to our department owing to suspected intramedullary tumors were registered and ultimately diagnosed with non-neoplastic intramedullary lesions, as described in the Materials and Methods section. Among the patients, 35/43 (81.4%) and 8/43 (18.6%) were eventually diagnosed as having neurological diseases and cervical myelopathy, respectively. To clarify our intention, the following text was modified:

(page 8, lines 274–278)

In the patients who were suspected of having an intramedullary tumor and referred to our department but finally diagnosed as having non-neoplastic spinal diseases, more than 80% were eventually diagnosed with neurological diseases. Meanwhile, others were diagnosed with compressive cervical myelopathy, such as cervical spondylotic myelopathy(CSM) and ossification of the posterior longitudinal ligament(OPLL).

The changes made to the revised manuscript are highlighted in red.

Reviewer 4-5) In the discussion section, line 205-206, the authors described, "However, notably, there was no case of spinal cord enlargement in astrocytomas in this study." However, the reviewer thinks that this was finding about non-neoplasticity. Please confirm.

  Thank you for highlighting this. The sentence was revised as follows:

(page 9 , lines 298-299)

  However, notably, all cases had spinal cord enlargement in astrocytomas in this study.

Round 2

Reviewer 1 Report

Overall this is a well written article on an often times difficult clinical topic.  It's difficult to say that this work will change the way these lesions are diagnosed, but I feel offers an adequate overview and would support publication

Reviewer 3 Report

I am satisfied with the revisions performed,

Thank You